# CCL24 and Fibrosis: A Narrative Review of Existing Evidence and Mechanisms

**DOI:** 10.3390/cells14020105

**Published:** 2025-01-13

**Authors:** Raanan Greenman, Chris J. Weston

**Affiliations:** 1Chemomab Therapeutics Ltd., Tel Aviv 6158002, Israel; 2Department of Immunology and Immunotherapy, School of Infection, Inflammation and Immunology, College of Medicine and Health, University of Birmingham, Birmingham B15 2TT, UK; 3National Institute for Health and Care Research (NIHR), Birmingham Biomedical Research Centre, Birmingham B15 2TT, UK

**Keywords:** CCL24, CCR3, chemokine, eotaxin-2, fibrosis, systemic sclerosis, metabolic dysfunction-associated steatohepatitis, primary sclerosing cholangitis

## Abstract

Tissue fibrosis results from a dysregulated and chronic wound healing response accompanied by chronic inflammation and angiogenesis. Regardless of the affected organ, fibrosis shares the following common hallmarks: the recruitment of immune cells, fibroblast activation/proliferation, and excessive extracellular matrix deposition. Chemokines play a pivotal role in initiating and advancing these fibrotic processes. CCL24 (eotaxin-2) is a chemokine secreted by immune cells and epithelial cells, which promotes the trafficking of immune cells and the activation of profibrotic cells through CCR3 receptor binding. Higher levels of CCL24 and CCR3 were found in the tissue and sera of patients with fibro-inflammatory diseases, including primary sclerosing cholangitis (PSC), systemic sclerosis (SSc), and metabolic dysfunction-associated steatohepatitis (MASH). This review delves into the intricate role of CCL24 in fibrotic diseases, highlighting its impact on fibrotic, immune, and vascular pathways. We focus on the preclinical and clinical evidence supporting the therapeutic potential of blocking CCL24 in diseases that involve excessive inflammation and fibrosis.

## 1. Introduction

Fibrosis, a leading cause of morbidity and mortality, results from a dysregulated wound healing process [1,2]. Wound healing under physiological conditions is marked by angiogenesis, fibroblast expansion, and re-epithelialization following the resolution of acute tissue injury. However, in chronic tissue injury caused by various factors such as autoimmune diseases, toxins, or pathogens, the body’s natural wound healing process becomes dysregulated [3]. The balance between extracellular matrix (ECM) synthesis and degradation shifts towards scar tissue accumulation. Therefore, aberrant wound healing and ongoing chronic injury can lead to fibrosis, tissue scarring (sclerosis), and the loss of organ function. Additionally, in oncological contexts, the dense fibrotic environment forms a physical barrier that impedes immune cell and drug penetration, thereby contributing to therapeutic resistance [4]. Overall, fibrosis represents a significant global health burden, affecting a quarter of the global population and causing major complications in nearly 5% of individuals annually [5].

Chemokines are 8–13 kDa secreted proteins which mediate the recruitment of immune cells to the site of injury, and are implicated in multiple inflammatory diseases such as atherosclerosis, multiple sclerosis, psoriasis, and insulin resistance [6]. Chemokines are categorized into four groups, CXC, CC, C, and CX3C, based on the number of amino acids between the N-terminal cysteine residues. They exert their diverse effects by interacting with G-protein-coupled receptors (GPCRs) on various target cells. Eotaxins are a subfamily of the CC chemokines, initially identified as potent eosinophil chemoattractants [7]. The eotaxin family is composed of three chemokines, CCL11 (eotaxin-1), CCL24 (eotaxin-2), and CCL26 (eotaxin-3), which bind to the same cognate receptor, CCR3 [7]. The function of CCL24 and CCL11 is conserved between humans and mice, and many preclinical models have elucidated their pleiotropic functions in a variety of diseases [8].

The current review focuses on the role of CCL24 in promoting inflammation and fibrosis, as well as the therapeutic potential of blocking CCL24 in diseases with extensive fibrosis. We will provide an overview of fibrosis mechanisms, discuss the general role of CCL24 in fibrotic processes, and subsequently delve into three representative fibro-inflammatory conditions, examining the preclinical and clinical evidence supporting the involvement of CCL24.

## 2. General Mechanisms of Fibrosis

### 2.1. Fibroblast Activation

Mechanistically, fibrosis occurs through the recruitment and activation of the following three main cell types: fibroblasts, immune cells, and endothelial cells [9]. Fibroblasts are the major cellular mediators of ECM remodeling, which under normal conditions are quiescent and are activated when tissue-repair mechanisms are initiated [10]. Several stimuli have been shown to trigger fibroblast activation, including transforming growth factor beta (TGFβ), wingless and Int-1 (WNT), platelet-derived growth factor (PDGF), fibroblast growth factor (FGF), mechanotransduction signaling via integrins, and damage-associated molecular patterns (DAMPs) [11,12,13].

The activation of fibroblasts involves their differentiation into myofibroblasts, which, through the overexpression of contractile proteins such as α-smooth muscle actin (α-SMA), can mechanically remodel the ECM via traction force [14]. Myofibroblasts also promote fibrosis and scarring by shifting the dynamic balance of ECM synthesis, cross-linking, and degradation [15,16]. Myofibroblasts exhibit an increased production of ECM proteins, particularly collagen [10]. Furthermore, activated fibroblasts promote enhanced tissue remodeling by increasing the expression of several ECM-modifying enzymes, such as matrix metalloproteinases (MMPs) and tissue inhibitors of metalloproteinases (TIMPs) [17,18,19]. Beyond the contraction and expression of ECM and ECM-modifying proteins, fibroblasts secrete a variety of cytokines, chemokines, adipokines, and growth factors that influence other cells, such as immune cells or tissue resident-epithelial and endothelial cells [10]. Fibroblasts can also become senescent, which may be beneficial during homeostasis, but can drive proinflammatory and profibrotic responses following prolonged periods of injury, potentially through a senescence-associated secretory phenotype (SASP) [20,21].

### 2.2. Immune Cell Recruitment

Inflammation and immune cell recruitment are essential in the initiation and progression of fibrosis [22]. Both innate and adaptive immune responses play a prominent role in the fibrotic process. The innate immune response is an early event of fibrosis that amplifies the process, and the adaptive immune response, predominantly lymphocytes, regulates the fibrosis towards a type 2 immune response [23,24]. The recruitment of immune cells and the activation of tissue-resident immune cells are induced by tissue injury- and inflammation-related factors, including pathogen-associated molecular patterns (PAMPs), DAMPs, proinflammatory cytokines, and adhesion molecules presented by the tissue endothelium [23,25,26,27]. Importantly, the recruitment of lymphocytes, monocytes, and neutrophils is essential in fibrosis progression [28,29,30,31,32]. Several chemokines are involved in the immune process and contribute to fibrosis progression. For instance, CCL2 (MCP-1) attracts monocytes which differentiate to macrophages in the tissue, contributing to the accumulation of macrophages in the fibrotic niche [33,34,35]. Macrophages are essential in tissue repair and fibrosis, contacting and activating different cell types in the fibrotic niche—including epithelial cells, endothelial cells, and fibroblasts—and releasing profibrotic cytokines and MMPs [36,37,38,39,40]. Macrophages have been categorized historically into classically (M1, proinflammatory) or alternatively (M2, anti-inflammatory and pro-fibrotic) activated types, although recent advances in RNA sequencing have identified multiple macrophage subsets which display a high degree of plasticity, and suggest ‘M1-like’ and ‘M2-like’ as the preferred terms for these populations [41,42]. Increased levels of M2-like macrophages have been observed in patients with fibrotic diseases where they secrete pro-fibrotic and pro-migratory factors, such as TGFβ, PDGF, and IL-6 [43,44,45]. Additionally, a highly conserved subpopulation of peripheral monocyte-derived macrophages, called scar-associated macrophages (SAMacs), has been found in fibrotic niches across various organs [46], where they secrete profibrogenic factors and promote fibrillar collagen expression by fibroblasts [47].

### 2.3. Endothelial Cell Activation

Although less studied, recent research has demonstrated that endothelial cells (ECs) and vascular remodeling are drivers of fibrosis development. Following injury, activated fibroblasts, macrophages, and damaged epithelial and endothelial cells secrete pro-angiogenic factors [48,49,50,51]. Additionally, chronic fibrosis and inflammatory cell accumulation create a hypoxic environment that activates hypoxia-inducible factors (HIFs), stimulating the production of proangiogenic factors [52]. This imbalance between angiogenic and angiostatic factors, favoring pro-angiogenic factors, contributes to the development of several fibrotic diseases [53,54,55]. Pro-angiogenic factors such as IL8, CXCL1, CXCL5, and VEGF aggravate the fibrosis of various organs, including the bones, liver, lungs, and skin [56,57,58,59,60,61,62,63]. Indeed, pulmonary fibrosis can be treated with Nintedanib, an inhibitor of the downstream signaling of VEGF, PDGF, and FGF, which interferes with angiogenesis, and Sorafenib, a multikinase inhibitor that blocks VEGF signaling, has been used to target angiogenesis in hepatocellular carcinoma [64,65,66]. ECs contribute to fibrosis progression by two mechanisms: signaling by injured/senescent ECs and the transition to a mesenchymal phenotype (EndMT) [67]. Endothelial cell injury-induced senescence, caused by tissue damage, promotes the production of senescence-associated secreted proteins, which may induce fibroblast activation, immune cell infiltration, and subsequent cytokine production, leading to tissue fibrosis [54,68,69]. Furthermore, senescent cells lower endothelial cell–cell and cell–matrix interactions, leading to reduced vascular barrier strength and increased immune cell adhesion [70]. During EndMT, major gene expression changes in ECs, driven by the transcription factor Snail1 [71], induce a phenotypic transformation to acquire a fibroblast-like phenotype, characterized by changes in morphology and migratory behavior. The transitioned cells are a source of myofibroblasts, produce ECM proteins, and secrete proinflammatory cytokines [72]. EndMT has been implicated in the development of fibrosis in various organs, including the intestine, liver, lungs, skin, kidneys, and heart [73,74,75,76,77,78].

## 3. CCL24 in Fibrotic Diseases

The CCR3 GPCR was shown to be involved in several fibrotic and inflammatory diseases [79,80,81,82,83]. Ligand binding to CCR3 triggers receptor dimerization/oligomerization, which subsequently activates classical G-protein-dependent or arrestin-dependent downstream signaling [84]. Several chemokines which are implicated in fibrosis can activate CCR3, including CCL3, CCL5, CCL7, CCL8, CCL11, CCL24, CCL26, and CCL28 [7,84]. Of these, the most potent CCR3 activators are the eotaxins [85]. Whereas all eotaxins are implicated in a variety of fibro-inflammatory diseases, and may have compensatory roles in knockout mice, each eotaxin shows a distinct profile of cell recruitment and activation, depending on the specific pathological conditions, their cellular origins, and the timing of expression following injury/damage [86,87,88,89,90,91]. For instance, CCL24 is the predominant eotaxin expressed by tumor-infiltrating eosinophils in Langerhans cell histiocytosis, and has a unique association with PSC-related pathways in the liver [88,89]. Surprisingly, even when applied to the exact same CCR3-expressing fibroblasts, CCL24, but not CCL26, stimulated proliferation and collagen synthesis [90]. Therefore, while many ligands can activate CCR3 to promote fibrosis and inflammation, the role of CCL24 is distinct in its effect, its localization, and its timing. Additionally, in contrast to other CCR3-activating chemokines, which typically exhibit promiscuous receptor binding, CCL24 exclusively activates CCR3, rendering it a promising therapeutic target for selective inhibition of this signaling pathway.

CCL24 promotes chemotaxis and activation of CCR3-expressing cells, including monocytes, neutrophils, basophils, T cells, fibroblasts, and eosinophils [92,93,94,95,96]. CCL24, produced by activated immune cells, primarily M2-like macrophages, and activated epithelial cells, is implicated in a variety of inflammatory, fibrotic, and vascular processes [97,98,99,100]. It induces and sustains a type 2 immune response by recruiting type 2-inducing cells, such as Th2, and by activating profibrogenic cells, including the polarization of macrophages towards an M2-like state [79,101,102]. CCL24 expression is upregulated by mechanical sensing and the type 2 cytokines IL4 and IL13, and can be further induced by IL10 [100,102,103]. As a type 2 cytokine, CCL24 was found to participate in the development of various diseases, amongst which are primary biliary cirrhosis, idiopathic pulmonary fibrosis, rheumatoid arthritis, atherosclerosis, asthma, as well as in malignancies [104,105,106,107,108,109]. This review will focus on the involvement of CCL24 in three fibrotic and inflammatory diseases: SSc, PSC, and MASH. Prominent preclinical and clinical studies of CCL24-blocking antibodies in these diseases are summarized in Table 1.

## 4. CCL24 in SSc

SSc is a rare autoimmune disease characterized by systemic inflammation and fibrosis, affecting various organs, including the skin, lungs, blood vessels, kidneys, heart, and digestive system [110]. The fibrotic process arises from vascular endothelial damage and immune system activation followed by autoantibody formation, leading to excessive extracellular matrix production [111,112]. SSc patients with complications of the lung, heart, or kidney exhibit a 3-year survival rate of 47–56%, which is notably low compared with patients with other connective tissue diseases [113,114,115]. Although two drugs are approved for the treatment of SSc-associated interstitial lung disease (ILD), no drugs limit the extent of cutaneous sclerosis [116].

SSc is initially detected by the presence of autoantibodies in the blood [117]. The autoimmune attack primarily targets blood vessels, especially ECs, leading to inflammation and EndMT, and results in blood vessel disintegration, impaired vascularization, and tissue damage [118]. EndMT increases the number of myofibroblasts, thus promoting fibrosis. EndMT is also implicated in SSc-associated ILD and pulmonary arterial hypertension (PAH) [119,120].

Elevated serum CCL24 levels are correlated with various stages of SSc and with higher SSc-related mortality rates, and both CCL24 and its cognate receptor, CCR3, are overexpressed in the skin of patients with SSc [54,121,122]. In the skin of patients with SSc, CCR3 is expressed on dermal fibroblasts, microvascular endothelial cells, and macrophages [121,123]. Moreover, CCL24 serum levels are associated with complications of PAH, ILD, telangiectasia, calcinosis, digital ulcers, and synovitis [122,124].

In-vivo models were used to demonstrate the role of CCL24 in SSc. Using bleomycin-induced models of dermal and pulmonary fibrosis, CCL24 knockout prevented dermal thickening, reduced collagen content, and reduced white blood cell and mononuclear cell counts in bronchoalveolar lavage (BAL) fluid [121]. Similarly, blocking CCL24 using a monoclonal antibody reduced collagen content and white blood cell counts in BAL fluid in the bleomycin-induced pulmonary fibrosis model [121], reinforcing a role for the CCL24-CCR3 axis in dermal and pulmonary inflammation, fibrosis, and vasculopathy. In-vitro, CCL24 induced the migration and myofibroblast-differentiation of dermal fibroblasts [121], and blockade of CCL24 reduced EC activation by SSc sera, EC angiogenesis, and prevented in-vitro-induced EndMT [97,121,124]. In pulmonary inflammation, CCR3 is expressed by lung fibroblasts, alveolar type II epithelial cells, and pulmonary infiltrating immune cells such as T cells, eosinophils, mast cells, neutrophils, and monocytes [90,125,126,127,128,129,130]. CCL24 was shown to induce inflammatory cell infiltration into BAL fluid in pulmonary inflammation, and to induce lung fibroblast proliferation and collagen synthesis [90,131]. Additionally, CCL24’s established role in cardiac fibrosis following injury suggests that CCL24 can promote SSc-associated cardiac fibrosis. CCL24 levels are elevated in cardiac remodeling and cardiac fibrosis following injury, and blocking CCL24 can prevent cardiac fibrosis [79]. Cardiac CCL24 is secreted by resident-macrophages and induces the activation and proliferation of cardiac fibroblasts [91]. Altogether, targeting CCL24 has the potential to interfere with inflammation, fibroblast activation, and vasculopathy in patients with SSc.

## 5. CCL24 in PSC

PSC is a chronic, progressive liver disease characterized by inflammation and fibrosis of intrahepatic or extrahepatic bile ducts, resulting in biliary strictures. This condition can lead to liver damage, cirrhosis, and, eventually, liver failure [132]. Patients with PSC are also at a high risk of developing biliary tract malignancies [132]. There is currently no FDA-approved drug for PSC, and treatment options are limited. Liver transplantation is the only effective treatment for PSC patients with decompensated cirrhosis [133]. Moreover, the recurrence after liver transplantation occurs in 9–27% of patients, increasing morbidity and mortality [134].

The pathogenesis of PSC remains unclear; however, it is evident that the initiation and progression of the disease is affected by abnormal periductular inflammation, fibrosis, and the activation of biliary epithelial cells (cholangiocytes, the epithelial cells lining the bile ducts). The activation of cholangiocytes, in response to an injury, is heterogeneous, leading to both a proliferative ductular reaction phenotype as well as cholangiocyte senescence and cell death [135]. The role of both autoimmunity and inflammation may be evidenced by the high occurrence of patients with PSC experiencing concomitant inflammatory bowel disease (IBD) or autoimmune hepatitis (AIH) [136,137]. The overlap with AIH and the limited response to immunosuppressants may indicate that, similarly to IBD, the inflammatory phase progresses to a fibrotic stage [137,138]. The chronic injury evident in PSC promotes common mechanisms of fibrosis, specifically by activating hepatic stellate cells (HSCs, liver resident precursors to myofibroblasts [139]) and portal myofibroblasts [140]. A key feature of biliary fibrosis in PSC is an “onion skin” of excessive ECM surrounding the injured bile duct, which is associated with biliary strictures and fibro-obliterative lesions.

In patients with PSC, CCL24 serum levels correlate with fibrotic markers, and distinctly higher CCL24 levels are observed in patients with cirrhosis [102,141]. A higher fraction of peripheral blood mononuclear cells (PBMCs) express CCR3 in patients with PSC compared to healthy controls [102], and both CCL24 and CCR3 are highly expressed in the liver of patients with PSC [102]. Periductular CCL24 is expressed by cholangiocytes and liver macrophages, whereas CCR3 is expressed by cholangiocytes, periductular immune cells, and α-SMA-expressing myofibroblasts.

Recently, clinical results were reported from a double-blind period of a phase 2 study in patients with PSC treated with a humanized anti-CCL24 monoclonal antibody (CM-101), NCT04595825 [142]. CM-101 exhibited anti-fibrotic, anti-inflammatory, and anti-cholestatic effects in patients with PSC, as evident by reductions in liver stiffness measurements, enhanced liver fibrosis (ELF) scores, pruritus, and serum markers of fibrosis and inflammation.

The mechanisms underlying the therapeutic benefit of CCL24 inhibition have been examined in several experimental models of PSC, including α-naphthylisothiocyanate (ANIT) diet, bile duct ligation (BDL), and knockout of Mdr2 gene [89,102]. CCL24 blockade therapy in Mdr2-knockout mice showed reductions in biliary hyperplasia, liver collagen content, liver inflammation score, and liver expression of profibrotic genes, accompanied by a dose-dependent reduction in serum ALT, ALP, and bile acids [102]. Similarly, ANIT and BDL models showed reduced liver fibrosis and reduced biliary hyperplasia following CCL24 blockade [89,102]. Proteomic analysis of sera from patients with PSC revealed that CCL24 is associated with pathways of leucocyte chemotaxis, HSC activation, and T cell activation [89]. Studying the underlying mechanisms of these effects revealed that CCL24 influences all of the key cell types involved in PSC pathogenesis: immune cells, fibroblasts, and cholangiocytes. CCL24-blocking antibody significantly inhibited CCL24-induced HSC proliferation, motility, and expression of proinflammatory and profibrotic genes (TIMP1, IL1β, α-SMA, and procollagen I) [89,102,143]. In addition to its effects on HSCs, CCL24 was found to induce cholangiocyte senescence and proliferation, promote M2-like macrophage polarization and proliferation, and, specifically, to recruit monocytes and neutrophils to the injured biliary area [89,102]. These two immune cell populations are important propagators of PSC activity, leading to periductular accumulation of monocyte-derived macrophages, especially M2-like macrophages, and neutrophils [144,145,146,147]. Targeting monocyte and neutrophil recruitment reduces liver inflammation and fibrosis [148,149,150]. Altogether, targeting CCL24 showed an effect on multiple biological processes involved in PSC pathogenesis, including inflammation, fibrosis, and cholestasis.

## 6. CCL24 in MASH

Metabolic dysfunction-associated steatotic liver disease (MASLD) is the most prevalent chronic liver disease, affecting millions worldwide [151]. It is characterized by the accumulation of fat in the liver (steatosis) and ranges from simple steatosis to MASH, a severe form associated with inflammation, fibrosis, and damage to hepatocytes. MASH can progress to cirrhosis and liver failure. MASH is the second leading cause of liver disease among adults awaiting liver transplantation, with an increasing prevalence worldwide [152,153]. Furthermore, MASH is an emerging risk factor for type 2 diabetes, cardiovascular disease, and end-stage kidney disease [151,154]. Liver fibrosis is prevalent in MASH and is closely related to disease progression, leading to cirrhosis and liver-related mortality [155]. Therefore, the level of fibrosis predicts disease progression and mortality [156,157]. Only one drug is currently approved for the treatment of MASH, which ameliorates liver fat content [158]. It is therefore crucial to identify new treatments that can prevent fibrosis progression.

Serum CCL24 levels are elevated in patients with MASLD compared to healthy individuals, with higher levels associated with more severe fibrosis [143]. PBMCs isolated from patients with MASLD have a higher expression of CCR3 [143]. CCL24 and CCR3 are also highly expressed in the livers of patients with MASH [143]; CCL24 colocalizes with immune cells, endothelial cells, hepatocytes, and cholangiocytes, whereas CCR3 colocalizes with endothelial cells, hepatocytes, and α-SMA-expressing myofibroblasts.

Two recent clinical studies evaluated the safety and activity of a humanized anti-CCL24 monoclonal antibody (CM-101) in patients with MASLD or with MASH [159,160]. CM-101-treated patients demonstrated moderate improvements in their fibrosis biomarkers, including serum ProC3 and TIMP1 levels (serum markers of ECM synthesis and turnover) and the enhanced liver fibrosis score (composite score based on the levels of circulating hyaluronic acid, procollagen III amino-terminal peptide (PIIINP), and TIMP1).

Blocking CCL24 was shown to reduce liver damage in various experimental animal models of MASH, including the methionine-choline-deficient (MCD) diet-induced and streptozotocin-high-fat-diet (STAM)-induced MASH models, and the thioacetamide (TAA)-induced liver fibrosis model [143]. These models demonstrated anti-inflammatory and anti-fibrotic effects following the attenuation of CCL24 function. These effects were evident by improved histologic scoring (decreased steatosis, inflammation, and hepatocyte ballooning), reflected in improved NAFLD activity scores (the histological assessment of MASLD), reduced liver collagen content (including a remarkable minimal visible fibrosis in rats receiving TAA and a CCL24-blocking antibody), reduced expression of liver inflammatory (IL6) and fibrotic genes (including TIMP1, Col1a1, and TGFβ), and reduced serum levels of liver enzymes and bilirubin. The anti-inflammatory and anti-fibrotic effects are thought to occur by blocking CCL24-induced chemotaxis of immune cells and by inhibiting the activation of HSCs, as in-vitro CCL24-blocking antibody significantly inhibited CCL24-induced HSC activation [89,102,143]. Similarly, blocking of the CCR3 receptor reduced macrophage and monocyte infiltration and collagen deposition in MASLD models [149,161]. These findings reinforce the role of CCL24 in liver fibrosis and highlight the potential value of inhibiting CCL24 to reduce liver inflammation, fibrosis, and overall liver damage.

## 7. Conclusions

Unbalanced inflammation and fibrosis, as manifested by the recruitment of immune cells and the activation of resident fibroblasts, epithelial cells, and endothelial cells, have been implicated in several diseases, including SSc and PSC. Recent research has uncovered a key role for the CCL24-CCR3 axis in the pathogenesis of these diseases. Specifically, CCL24 has emerged as a key driver of both inflammation and fibrosis. Blocking CCL24-associated signaling pathways represents a novel and promising therapeutic strategy for treating a wide range of fibrotic diseases. A first-in-class CCL24-blocking antibody (CM-101) has already shown promising results in both preclinical and clinical studies of SSc, PSC, and MASH. The ongoing research efforts, focused on CCL24 and its inhibition, hold immense potential for transforming the treatment landscape for fibrotic diseases, offering hope for improved outcomes for patients worldwide.

**Table 1 cells-14-00105-t001:** Preclinical and clinical studies examining CCL24 blockade in SSc, PSC, and MASH.

Animal Model/Indication	Anti-CCL24 mAb Treatment	Treatment Effect	Reference/NCT
**Preclinical in-vivo models**			
Intraperitoneal 5 μg CCL24 injection to BALB/C male mice followed by characterization of immune cell recruitment	100 μg subcutaneous injection.	Reduced infiltration of monocytes and neutrophils; reduced M2-like macrophage accumulation.	[89]
Bleomycin subcutaneous injection model of SSc in C3H female mice	Intraperitoneal injection of 0.5–2.5 mg/kg	Reduction in dermal thickness, skin collagen, and immune cell infiltration to the lungs.	[121]
Bleomycin intra-tracheal injection model of SSc in C57BL/6 male mice	Intraperitoneal injection of 2.5 mg/kg	Reduction in lung collagen and immune cell infiltration to the lungs.	[121]
Experimental liver cholangitis spontaneously induced in MDR2 knockout mice	Subcutaneous 5 mg/kg injection, twice weekly	Reduced serum liver enzymes and bile acid levels, cholangiocyte proliferation, liver fibrosis, liver inflammation, and periductular macrophage accumulation.	[102]
TAA-induced liver fibrosis model in Wistar rats	Intravenous 2.5 mg/kg, twice weekly	Reduced serum liver enzymes, liver collagen concentration, and fibrosis.	[143]
Liver fibrosis model induced by bile duct ligation in Sprague Dawley rats	Intravenous 10 mg/kg, twice weekly	Reduced liver collagen content and cholangiocyte proliferation.	[102]
ANIT-induced cholestasis mouse model in male C57BL/6 mice	Intraperitoneal injection of 5 mg/kg, twice weekly	Reduction in bile acid and histopathology damage.	[89]
TAA-induced liver fibrosis model in mice	Subcutaneous 2.5 mg/kg and 10 mg/kg, twice weekly	Reduced serum liver enzymes, liver inflammation, and liver fibrosis.	[162]
MCD diet-induced MASH mice model in male and female C57BL/6 mice	Intraperitoneal 5 mg/kg, twice weekly	Reduced serum liver enzymes and bilirubin, liver steatosis, liver inflammation, and NAFLD score.	[143]
STAM-induced MASH model in C57BL/6 mice	Intraperitoneal 5 or 7.5 mg/kg injection, twice weekly	Reduced NAFLD score, liver collagen concentration, liver steatosis, and liver inflammation.	[143]
**Clinical studies**			
Evaluation of the safety, tolerability, PK, and PD in subjects with MASLD	5 doses administered every 3 weeks; 2.5 mg/kg intravenous or 5 mg/kg subcutaneous infusions of anti-CCL24 mAb (CM-101) or placebo	CM-101 was safe and well tolerated. Improvement in inflammatory and fibrotic markers.	[159] NCT06044467
Evaluation of the mechanism of action, safety, tolerability, PK, and PD in subjects with MASH	8 doses administered every 2 weeks; 5 mg/kg subcutaneous infusions of anti-CCL24 mAb (CM-101) or placebo	CM-101 was safe and well tolerated. Improvement in inflammatory and fibrotic markers.	[160] NCT05824156
Evaluation of the mechanism of action, safety, tolerability, PK, and PD in subjects with PSC	5 doses administered every 3 weeks; 10 or 20 mg/kg intravenous infusions of anti-CCL24 mAb (CM-101) or placebo	CM-101 was safe and well tolerated. Improvement in fibrotic markers. Improvement in cholestatic markers. Improvement in inflammatory markers.	[142] NCT04595825

## Data Availability

No new data were created or analyzed in this study.

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
