# Peer review of "CCL24 and Fibrosis: A Narrative Review of Existing Evidence and Mechanisms"

_cells, 2025, doi:10.3390/cells14020105_

Round 1
Reviewer 1 Report
Comments and Suggestions for Authors
Dear Authors,
Serious, large review giving the responses to the asked questions. However, I have some small objections for this manuscript:
1) please, add also the plan of the review at the end of Section 1; additionally, develop please one subsection (methodological one) where you indicate the data bases used, time (from...to), key words, inclusion, also exclusion creteria for the manuscripts;
2) Table 1, whats included as the conclusions, sorry, doesnt fit in this place. I would like to ask you to move it as separate subsection before the conclusions with short description and title (what is missed now...);
3) Conclusions will be as a section 8 then. Well, extraordinar a little bit, but actually I liked the scheme here. Just add please abbreviations at the end of title here;
4) Outstanding References from the point of view in number of them. However, carelessness gives some unpretty impression, sorry... So, not finished reference (100); not correctly arranged others - 74, 123, 119, 139... Please, go through carefully and keed the arrangement of References in the accordance to the requirements of the Journal! Additionally, think about these 4 previous century sources what you have mentioned in the manuscript - do you really need them. Because somehow do not fit to otherwise well written and modern manuscript!
Author Response
Comments 1: please, add also the plan of the review at the end of Section 1; additionally, develop please one subsection (methodological one) where you indicate the data bases used, time (from...to), key words, inclusion, also exclusion creteria for the manuscripts;
Response 1: A plan of the review was added at the end of section 1: “We will provide an overview of fibrosis mechanisms, discuss the general role of CCL24 in fibrotic processes, and subsequently delve into three representative fibro-inflammatory conditions, examining preclinical and clinical evidence supporting the involvement of CCL24.” Lines 55-58
Given this paper's structure as a narrative review, rather than a scoping or systematic review, a methodology section is not applicable. To prevent confusion, the word “narrative” was added to the paper’s title. Line 2.
Comments 2: Table 1, whats included as the conclusions, sorry, doesnt fit in this place. I would like to ask you to move it as separate subsection before the conclusions with short description and title (what is missed now...);
Response 2: Table 1 is now referenced at the end of section 3 (instead of section 7), hence there is no need to add a new section. “Prominent preclinical and clinical studies of CCL24-blocking antibodies in these diseases are summarized in Table 1.” Lines 171-173
A title was added to Table 1.
Comments 3: Conclusions will be as a section 8 then. Well, extraordinar a little bit, but actually I liked the scheme here. Just add please abbreviations at the end of title here;
Response 3: We revised the manuscript so that the scheme will turn to a graphical abstract instead of being Figure 1, and is not referenced in section 7 (conclusions).
Abbreviations were added to the legend of the scheme.
Comments 4: Outstanding References from the point of view in number of them. However, carelessness gives some unpretty impression, sorry... So, not finished reference (100); not correctly arranged others - 74, 123, 119, 139... Please, go through carefully and keed the arrangement of References in the accordance to the requirements of the Journal! Additionally, think about these 4 previous century sources what you have mentioned in the manuscript - do you really need them. Because somehow do not fit to otherwise well written and modern manuscript!
Response 4: We use Zotero reference software, which has the MDPI formatting style installed within it.
Whenever possible, formatting was corrected. Some references are derived from conferences and are therefore in the format of the abstract book. If the revised bibliography doesn’t match MDPI requirements, specific un-matched references can be altered manually.
Papers from the 90’ were replaced by newer ones.
Reviewer 2 Report
Comments and Suggestions for Authors
In this review article, the authors have well summarized the pathological roles and therapeutic potential of CCL24 in tissue fibrosis, along with elaborating on the general process of tissue fibrosis. However, some important information is missing or needs to be explained in more detail. In order to improve the manuscript, I would like the authors to address the following concerns.
1. lines 110–112: Some examples of specific cases, in which pro-angiogenic factors promote fibrosis, should be given.
2. How is the imbalance caused?
3. lines 126–145: The reasons to omit CCL11 and CCL26 are unclear and should be explained in more detail. The involvement of both in the development of fibrosis-related diseases has been reported in many studies.
4. CCR3 binds to not only CCL11/24/16, but also other chemokines (Cancers 2020:12:1383). It should be discussed whether CCL24 is a sole factors that activates CCR3 in the development of fibrosis.
5. Is it possible that CCL24 has a CCR3-independent function?
6. If relevant reports are available, the molecular mechanisms underlying the upregulation of CCL24 and/or CCR3 expression should be discussed.
Author Response
- Comments 1: lines 110–112: Some examples of specific cases, in which pro-angiogenic factors promote fibrosis, should be given.
Response 1: We thank the reviewer for stressing the need for more background and samples of the involvement of pro-angiogenic factors. This information was added to section 2: “Pro-angiogenic factors such as IL8, CXCL1, CXCL5 and VEGF aggravate fibrosis of various organs, including the bone, liver, lung and skin [56–63]. Indeed, pulmonary fibrosis can be treated with Nintedanib, an inhibitor of downstream signaling of VEGF, PDGF and FGF, which interferes with angiogenesis, and Sorafenib, a multikinase inhibitor that blocks VEGF signalling, has been used to target angiogenesis in hepatocellular carcinoma [64–66]. ”. Lines 119-124
comments 2: How is the imbalance caused?
Response 2: A description of cells and conditions that shift the balance was added to section 2: “Following injury, activated fibroblasts, macrophages, and damaged epithelial and endothelial cells secrete pro-angiogenic factors [48–51]. Additionally, chronic fibrosis and inflammatory cell accumulation create a hypoxic environment that activates hypoxia-inducible factors (HIF), stimulating the production of proangiogenic factors [52].” Lines 113-117.
comments 3: lines 126–145: The reasons to omit CCL11 and CCL26 are unclear and should be explained in more detail. The involvement of both in the development of fibrosis-related diseases has been reported in many studies.
Response 3: We appreciate the reviewer’s emphasis on adding some information on other eotaxins and CCR3-ligands. We therefore rearranged section 3 to discuss CCR3 in fibrosis, the involvement of other chemokines, and then focus on CCL24.
“The CCR3 GPCR was shown to be involved in several fibrotic and inflammatory diseases [78–82]. Ligand binding to CCR3 triggers receptor dimerization/oligomerization, which subsequently activates classical G protein-dependent or arrestin-dependent downstream signaling [83]. Several chemokines which are implicated in fibrosis can activate CCR3, including CCL3, CCL5, CCL7, CCL8, CCL11, CCL24, CCL26 and CCL28 [7,83]. Of these, the most potent CCR3 activators are the eotaxins [84]. Whereas all eotaxins are implicated in a variety of fibro-inflammatory diseases and may have compensatory roles in knockout mice, each eotaxin shows a distinct profile of cell recruitment and activation, depending on specific pathological conditions, their cellular origins and the timing of expression following injury/damage [85–89]. For instance, CCL24 is the predominant eotaxin expressed by tumor-infiltrating eosinophils in Langerhans cell histiocytosis, and has a unique association with PSC-related pathways in the liver [87,88]. Surprisingly, even when applied to the exact same CCR3-expressing fibroblasts, CCL24, but not CCL26, stimulated proliferation and collagen synthesis [89]. Therefore, while many ligands can activate CCR3 to promote fibrosis and inflammation, the role of CCL24 is distinct in its effect, its localization and its timing. Additionally, in contrast to other CCR3-activating chemokines, which typically exhibit promiscuous receptor binding, CCL24 exclusively activates CCR3, rendering it a promising therapeutic target for selective inhibition of this signaling pathway.
CCL24 promotes chemotaxis and activation of CCR3-expressing cells, including monocytes, neutrophils, basophils, T cells, fibroblasts, and eosinophils [90–94]. CCL24, produced by activated immune cells, primarily M2-like macrophages, and activated epithelial cells, is implicated in a variety of inflammatory, fibrotic, and vascular processes [95–98]. It induces and sustains a type 2 immune response, by recruiting type 2 inducing cells, such as Th2, and by activating profibrogenic cells, including the polarization of macrophages towards an M2-like state [99–101]. CCL24 expression is upregulated by mechanical sensing and the type 2 cytokines IL4 and IL13, and can be further induced by IL10 [98,101,102]. As a type 2 cytokine, CCL24 was found to participate in the development of various diseases, amongst which are primary biliary cirrhosis, idiopathic pulmonary fibrosis, rheumatoid arthritis, atherosclerosis, asthma, as well as in malignancies [103–108]. This review will focus on the involvement of CCL24 in three fibrotic and inflammatory diseases: SSc, PSC and MASH. Prominent preclinical and clinical studies of CCL24-blocking antibodies in these diseases are summarized in Table 1.” Lines 139-173.
Comments 4: CCR3 binds to not only CCL11/24/16, but also other chemokines (Cancers 2020:12:1383). It should be discussed whether CCL24 is a sole factors that activates CCR3 in the development of fibrosis.
Response 4: See reply to comment 3.
Comments 5: Is it possible that CCL24 has a CCR3-independent function?
Response 5: As far as we are aware, no paper has thus far demonstrated a CCR3-independent function. Internal unpublished results from Chemomab did not find binding to other receptors.
comments 6: If relevant reports are available, the molecular mechanisms underlying the upregulation of CCL24 and/or CCR3 expression should be discussed.
Response 6: Section 3 now includes the fibrotic factors as IL4 that induce CCL24 expression: “CCL24 expression is upregulated by mechanical sensing and the type 2 cytokines IL4 and IL13, and can be further induced by IL10”. Lines 165-167.
Reviewer 3 Report
Comments and Suggestions for Authors
T
This review article summarizes the relationship between CCL24 and fibrotic diseases (PSC, SSC, MASH, etc.) in parallel, citing many papers.
However, there is a lack of basic information about what kind of protein CCL24 is, how gene expression of CCL12 is regulated, and what kind of signal transduction it acts through. In addition, the article is too specialized in CCL24, and the description of the cross-talk with other important chemokines and cytokines involved in fibrosis is not sufficient.
For these reasons, it is not considered suitable as a review for a basic science journal.
Author Response
Comments 1: This review article summarizes the relationship between CCL24 and fibrotic diseases (PSC, SSC, MASH, etc.) in parallel, citing many papers.
However, there is a lack of basic information about what kind of protein CCL24 is, how gene expression of CCL12 is regulated, and what kind of signal transduction it acts through. In addition, the article is too specialized in CCL24, and the description of the cross-talk with other important chemokines and cytokines involved in fibrosis is not sufficient.
For these reasons, it is not considered suitable as a review for a basic science journal.
Response 1: We appreciate the reviewer’s emphasis on adding more background and relevant basic information to the paper. We addressed all these points by describing the factors that regulate CCL24 expression, the signal transduction through CCR3, the involvement of other CCR3 ligands in fibrosis and inflammation and compared CCL24 to them, and described the involvement of other important cytokines (namely, TGFb, PDGF, FGF, CCL2, CXCL1, CXCL5, CXCL8 and VEGF). We believe these additions significantly enhance the clarity and comprehensiveness of the manuscript.
Round 2
Reviewer 3 Report
Comments and Suggestions for Authors
This manuscript has been revised well.